# Awareness, Acceptance, and Associated Factors of Human Papillomavirus Vaccine among Parents of Daughters in Hadiya Zone, Southern Ethiopia: A Cross-Sectional Study

**DOI:** 10.3390/vaccines10121988

**Published:** 2022-11-23

**Authors:** Yilma Markos Larebo, Legesse Tesfaye Elilo, Desta Erkalo Abame, Denebo Ersulo Akiso, Solomon Gebre Bawore, Abebe Alemu Anshebo, Natarajan Gopalan

**Affiliations:** 1Department of Epidemiology, College of Medicine and Health Sciences, Wachemo University, Hossana P.O. Box 667, Ethiopia; 2Department of Public Health, College of Medicine and Health Sciences, Wachemo University, Hossana P.O. Box 667, Ethiopia; 3Department of Medical Laboratory Science, College of Medicine and Health Sciences, Wachemo University, Hossana P.O. Box 667, Ethiopia; 4Department of Midwifery, College of Medicine and Health Sciences, Wachemo University, Hossana P.O. Box 667, Ethiopia; 5Department of Epidemiology and Public Health, School Life Science, Central University of Tamil Nadu, Thiruvarur P.O. Box 610005, India

**Keywords:** Human Papilloma Virus, vaccine, awareness, parents, daughters, Ethiopia

## Abstract

Introduction: Human papillomavirus infections are the most prevalent sexually transmitted disease among women worldwide. Cervical cancer is the second-most frequent disease worldwide in terms of incidence and mortality, and it is primarily responsible for fatalities in low- to middle-income nations, including Ethiopia. Objective: To assess awareness, acceptance, and associated factors of the human papillomavirus vaccine among parents of daughters in the Hadiya zone, southern Ethiopia. Methods: From November to December 2021, a community-based cross-sectional study was conducted in the Hadiya zone among parents with daughters in the zone. The study respondents were chosen using a two-stage sampling technique from parents with a 9–14-year-old daughter. An interviewer-administered questionnaire was used to collect data. For analysis, the data were entered into Epidata version 3.1 and exported to SPSS version 25. Variables with a *p*-value less than 0.25 in the bivariate analysis were transferred to multivariable analysis. A logistic regression model was applied to forecast the association between the predictor and outcome variables. Statistical significance was considered at a 0.05 *p*-value. Results: The study showed that the overall acceptance of parents to vaccinate their daughters with HPV vaccination was 450 (84.9%). Parents of daughters of male sex (AOR: 0.407; 95%CI: 0.221, 0.748), who had only one daughter (AOR: 2.122; 95%CI: 1.221, 3.685), whose daughter(s) attended a government school (AOR: 0.476; 95%CI: 0.263, 0.861), who had poor knowledge (AOR: 0.532; 95%CI: 0.293, 0.969) and who had a negative attitude (AOR: 0.540; 95%CI: 0.299, 0.977) were discovered to have a strong correlation. Conclusion: This study found that there was a high level of parental acceptance; attitudes and knowledge about the HPV vaccine are significant in determining their intentions to vaccinate their daughter. Authorities in high-risk areas for cervical cancer incidence should plan and implement strategies by providing health information regarding human papillomavirus vaccination with an emphasis on raising community awareness.

## 1. Introduction

Human Papilloma Virus (HPV) is the name given to a group of viruses that includes more than 200 related viruses, more than 40 of which are spread through vaginal and anal intercourse and can cause premalignant changes and malignant cancers of the cervix, vagina, vulva, and anus [1]. In both men and women, HPV infections are the most common sexually transmitted illness (STI) [1,2]. If an HPV infection goes untreated without the removal of precursor lesions, it can lead to cancer [1,3].

Cervical cancer is the world’s most common cancer in women; every year, approximately 500,000 new cases emerge [2], accounting for 7.5% (270,000) of the leading causes of all female cancer death in women worldwide [4] with a malignant tumor arising from the cells of the uterine cervix [4]. It affects over a million women in the developed world, with HPV types 16 and 18 accounting for up to 70% of cases [5]. Sub-Saharan Africa (SSA) has the highest incidence and mortality rates in the world, accounting for more than 70% of the global cervical cancer burden [4].

In developing countries, the burden of 230,000 cases and deaths (80% of the total) is borne by those who have only the bare minimum of resources to deal with the situation [6]; despite being largely preventable, it is the second-most common cancer in developing countries in terms of incidence and mortality [7].

Cervical cancer is the second leading cause of cancer death among women in Ethiopia. Approximately 7600 new cervical cancer cases are diagnosed each year, with over 6000 deaths from the disease [2]. HPV is a major risk factor leading to the development of cervical cancer [1] and cervical cancer is caused by a persistent infection with one or more of the high-risk types of HPV [8]. It is the most common infection acquired during sexual relations, typically during adolescence [9], and 5% of those infected are women living with human immune virus (HIV) [10].

Cervical cancer screening is the most effective way for all women to reduce cancer mortality [10]. Although the majority of developed countries have well-organized screening, early detection, and effective treatment strategies for precancerous cervical lesions [4], more than 80% of sexually active women have HPV at some point in their lives [2].

As a result, the World Health Organization (WHO) has developed comprehensive strategies for cervical cancer prevention and control, including HPV vaccination as the primary method of prevention [5,11]. Cervical cancer prevention programs, such as HPV vaccination and cytology-based cervical cancer screening programs, have helped to reduce the incidence of the disease in developed countries [2].

Ethiopia introduced the HPV vaccine in 2018 with the help of the Global Alliance for Vaccine and Immunization (GAVI) [12], which targeted 14-year-old girls, and vaccines were administered through schools in two doses six months apart [13], but misconceptions about the cause and prevention of cervical cancer are common in Ethiopia due to a lack of awareness. As a result, appropriate studies to raise awareness, acceptance, and the use of services such as immunization must be conducted before the nationwide scaling-up of cervical cancer prevention programs can take place [2].

There is now an HPV prophylactic vaccine available that protects against most genital warts and cervical cancer. It is also known as the cervical cancer vaccine [1]. It has accounted for approximately 90% of cervical cancer prevention cases and 10% of other HPV-related cancers or diseases [13]. HPV vaccines are generally regarded as safe, with only minor and transient side effects, with a sore arm from the shot being the commonest, as well as headache, dizziness, nausea, fatigue, soreness, redness, or swelling at the injection site [14].

The WHO recommends two doses of the HPV vaccine for girls aged 9–14 years, and three doses for girls aged 15 years and older, as well as those who are immunocompromised or HIV positive (regardless of age) [11]. HPV vaccines work best for women who have never been exposed to HPV infection and are thus recommended for girls of appropriate ages before sexual debut; however, the vaccines do not prevent other sexually transmitted diseases or treat existing HPV infections or HPV-caused diseases [12]. It reduces/prevents infection with the following nine HPV types: HPV types 6 and 11, which cause 90% of genital warts; HPV 16, by far the most common type, accounting for 90–95% of oropharyngeal cancer; HPV types 16 and 18, which cause approximately 70% of cervical cancers [4]; and HPV types 31, 33, 45, 52, and 58, which cause 10% to 20% of cervical cancers [15,16].

Ethiopia had planned to introduce the HPV vaccine through a routine immunization program for approximately 6 million girls aged 9 to 14 years old [12,17]. However, due to a global shortage of the HPV vaccine, the country is introducing the vaccine in a single age cohort (14-year-old girls) in the first year and hopes to expand the introduction to additional age cohorts in the second year and beyond, based on global vaccine availability. If the vaccine shortage persists, the country will continue to vaccinate 14-year-old girls every year [17].

Basic knowledge of women’s pelvic anatomy and the natural history of cervical cancer provides primary- and secondary-level healthcare providers with the knowledge they need to effectively communicate and raise awareness of cervical cancer prevention in women, families, and communities [7].

According to the 2005 CSA report, the Hadiya zone, Gurage, and Kembata Tembaro produced 8364.00 tons of coffee in 2005, accounting for 8.33% of Southern Nation Nationality People Region (SNNPR) output and 3.36% of Ethiopia’s overall output. According to the World Bank report, only 6% of Hadiya people live with electricity, there is a road density of 104.1 km per 1000 square kilometers compared to the national average of 30 km, 0.6 hectares of land for rural households compared to the national average of 1.01 hectares of land, and an average of 0.89 for the SNNPR. There is the equivalent of 0.6 head of cattle; 22.8% of the population is working in non-farm related jobs, compared to 25% nationally and 32% regionally. The average age of first childbirth in the Hadiya zone is 18 years, which is similar to national figures, and the majority of the residents are Christian [18].

There are 376 health institutions in the zone, including 1 general hospital, 3 primary hospitals, 3 primary hospitals (under construction), 61 health centers, 311 health posts, 81 private clinics (1 higher, 16 medium, and 64 lower), and 39 private pharmacies (2 pharmacies, 17 drug stores, and 20 rural drug vendors) that provide routine health services to the community. Health coverage has not yet been achieved, and no health facilities presently provide immunization and HPV testing services to adolescents [19,20,21].

Even if the study location is known to the investigators, there is little information about awareness, acceptance, and associated factors of the human papillomavirus vaccine among parents of daughters in the Hadiya zone, southern Ethiopia. Therefore, this study aimed to determine awareness, acceptance, and associated factors of the human papillomavirus vaccine among parents of daughters in the Hadiya zone, southern Ethiopia.

## 2. Methods and Materials

### 2.1. Study Setting, Design, and Period

A community-based cross-sectional study design was conducted in the Hadiya zone among parents and daughters of the zone, from November to December 2021. The zone is located in the southwest of Ethiopia, 232 km away from Addis Ababa (the capital city of Ethiopia), and 194 km from Hawassa (the regional capital city) [19,21].

Administratively, the Hadiya zone is organized by 4 administrative towns, 13 districts, 305 rural kebeles, and 30 urban kebeles (the smallest administrative unit in Ethiopia), with an estimated population size of 1,727,920, with 856,357 (49.56%) males and 871,563 (50.44%) females [19,20,21].

### 2.2. Sample Size Determination

The sample size was calculated using the single population proportion formula, considering the following assumptions and taking a prevalence of 79.5%, which was a study conducted in the Bench-Sheko zone, southwest Ethiopia [2]: $n\;=\frac{{\left(Z\alpha /2\right)}^{2}p\;\left(1-p\right)}{d^{2}}$, where n = the desired sample size, *p* = parents’ acceptance to vaccinate their daughter = 79.5% (which was taken from a study conducted in the Bench-Sheko zone, southwest Ethiopia, 2021), Z1 − α/2 = critical value at 95% confidence level (1.96), d = the margin of error = 5%, n = (1.96)^2^. 0.795(1 − 0.795)/(0.05)^2^ = 250 and using design effect (Deff = 2), because a two-stage sampling technique was used, the final sample size required was 2 ∗ 250 = 500; for possible zero response during the study, the final sample size was increased by 10% to n = 500 + 10% of 500 which is 50; by adding that, the total sample size was 550.

### 2.3. Sampling Procedure

A multistage stage stratified sampling technique was used to select study respondents. In the first stage of sampling, districts from the Hadiya zone like Lemo, Misha, Soro, and Shashogo, as well as the Hossana and Shone town administrations, were chosen by lottery methods. In the second stage of sampling, kebeles were stratified as urban or rural, and two urban and ten rural kebeles were chosen by lottery methods. Then, thirty percent of kebeles from each of the selected districts were chosen.

The calculated samples were then distributed to each of the selected schools in kebeles under the probability proportional to the sample size allocation (PPS) principle. Respondents were caregivers of a daughter aged 9 to 14, such as parents, grandparents, or anyone else who had self-identified as being responsible for the daughter’s care. Male caregivers were interviewed if a female caregiver was unavailable. Female caregivers were preferred. If a household had more than one female between the ages of 9 and 14, one was chosen at random as the index female for interview questions. Finally, the 550 daughters were chosen using a systematic random sampling technique with a skip interval of every ‘5th’. As a result, households with daughters aged 9–14 were interviewed.

Parents of daughters aged ≥ 18 years old who live permanently in the study area (for more than 6 months) were included in the study. Those respondents who were unable to provide the necessary information, pregnant girls, girls who had recently given birth and were breastfeeding, and girls who were not attending classes despite being enrolled, were also excluded, and could not be recruited. When no person could participate in the study, the next household was used in their place.

### 2.4. Measurement of Variables

The purpose of this study was to determine awareness, acceptance, and associated factors of the human papillomavirus vaccine among parents of daughters. The dependent variable in this study was acceptance of the human papillomavirus vaccine. The predictor variables were respondents’ basic socio-demographic characteristics, such as age, sex, educational status, marital status, occupational status, monthly income level, knowledge, attitude, and awareness of HPV vaccination.

In this study, the knowledge of the parents of daughters on acceptance of the human papillomavirus vaccine was measured with a 17-response yes-or-no questionnaire. A correct response was assigned a value of one, while an incorrect response was assigned a value of zero. The scores for each item were tallied, and those who scored higher than the mean value were classified as having good knowledge, while those who scored lower were classified as having poor knowledge [2,3,22].

The attitude of the parents of daughters on acceptance of the human papillomavirus vaccine was determined by eleven Likert scale items ranging from strongly disagree to strongly agree. Those who scored higher than the mean value were classified as having a positive attitude, while those who scored lower were classified as having a negative attitude [2,3,22].

The acceptance of the parents of their daughter’s inoculation with the human papillomavirus vaccine was determined by the question “are you willing to vaccinate your daughter for HPV vaccination, which can protect against HPV infection?” (Choices: 1 as Yes or 0 as No) [2, 3, 22].

The awareness of the parents of daughters on acceptance of the human papillomavirus vaccine was measured by the question “today, had you heard of HPV?” Parents of daughters who answered ‘yes’ to this question were regarded to be aware of HPV [2,3,22].

### 2.5. Data Collection and Quality Assurance Procedures

The data were collected using interviewer-administered quantitative data. The questionnaire contained questions developed and adapted from various kinds of literature. The necessary permissions were obtained from the owner of the original questionnaire. The questionnaire developed by the investigators contained the following 4 sections: (1) basic demographic characteristic (age, sex, educational status, marital status, occupational status, and monthly income level), (2) knowledge, (3) attitude, and (4) information-related factors on vaccination awareness and acceptance of parents of daughters on acceptance of human papillomavirus vaccine. Three days of training were provided for data collection using an interview-administered method by two trained data collectors and one supervisor. The principal investigator reviewed the obtained data to ensure its accuracy, completeness, clarity, and consistency. The questionnaire was written in English, translated into common Amharic to ensure that the items were consistent, and then translated back into English to ensure that the common Amharic was accurate. The questionnaire’s face validity was established.

A pre-test was performed before data collection by taking 28 samples out of the study area. The data collection tool’s reliability was evaluated in terms of knowledge, attitude, and acceptance, as well as awareness. To ensure the quality of the data, emphasis was placed on the design of the data collection instrument for its simplicity and standardized community rating scales, validity, and reliability were considered.

### 2.6. Data Processing and Analysis

Data that had been coded was input in EpiData version 3.1 and exported to SPSS version 25 for analysis. Data entry fell under the purview of the principal investigator. Tables, graphs, and charts were used to perform descriptive analysis and report the results in frequency. Variables from the bivariate analysis that had *p*-values under 0.25 were added to the multivariable analysis.

To choose variables for multivariable analysis, bivariate analysis was used. However, multivariable statistical significance was tested at the level of 5%. Using logistic regression, adjusted odds ratios (AOR) and a 95% confidence interval were utilized to confirm the presence and strength of the link between the independent and dependent variables. The Hosmer and Lemeshow tests were used to gauge the model’s fitness. The Variance Inflation Factor (VIF), standard error, and correlation coefficient were used to test for multicollinearity among independently associated variables.

## 3. Results

### 3.1. Characteristics of Respondents

Out of the 550 parents of daughters who were eligible to participate in the study, 20 were excluded (20 data were incomplete and were not considered for analysis), resulting in a 96.54% response rate. Out of the 530 parents of daughters included in the study, the majority, 323 (60.9%) of the respondents were female, and almost more than half (323 (60.9%)) of them were in the age category 30–39 years. The mean age of parents of daughters was 38 with (SD) ± (9.45) years and nearly less than three-fourths, 365 (68.9%) of the respondents were Protestant, 354 (66.8%) were urban by residence, 437 (82.5%) were married, 322 (60.8%) had an educational status of degree and above, 354 (66.8%) were civil servants by occupation, and 389 (73.4%) were of Hadiya ethnicity. The majority of the respondents (418 (79.8%)) had more than one child, 355 (67%) respondents were educated in a private school, more than half of respondents, 318 (60%) had information on HPV, and 144 (45.3%) listed the major source of information as television or radio. Almost half, 291 (54.9%) of the respondents reported a monthly income >ETB 4999 (>USD 95.24) (Table 1).

Of the 530 study respondents, the majority 419 (79.1%) with 95% CI [75.5, 82.5] had good knowledge with a mean of 1.7906 and standard deviation of ±0.40729 (Figure 1).

Almost more than three-fourths 418 (78.9%) and nearly all most all 499 (94.2%) of the parents of daughters had ever heard about cervical cancer and thought cervical cancer infected only females, respectively. Nearly less than one-fourth 111 (20.9%) and 144 (27.2%) of the parents of daughters knew that cervical cancer produces no signs and symptoms at an early stage and that it is a fast-growing cancer, respectively. Concerning treatment curability, about 418 (78.9%) respondents answered the correct answers about early detection for cervical cancer protection and 371 (70%) respondents were getting a Pap test for early detection of cervical cancer. Almost three-fourths 387 (73%) respondents knew that HPV is the main cause of cervical cancer and the majority 435 (82.1%) knew it was common in women younger than 30 years old. More than three-fourths of respondents knew that HPV vaccination is available for girls aged 9 to 14 years and that cervical cancer risk can be reduced by HPV vaccination 419 (79.1%) and 434 (81.9%), respectively). Almost more than half of respondents knew that HPV infection is a sexually transmitted infection, and that vaccination should be received before sexual debut 307 (57.9%) and 338 (63.8%), respectively. The majority 434 (81.9%) and nearly less than one-fourth 112 (21.1%) of the respondents knew that the persistent infection of high-risk HPV could lead to cervical cancer and 70% of cervical cancer is caused by HPV types 16 and 18, respectively. More than one-fourth 159 (30%) of the parents of daughters were aware of the most common and effective treatment to cure cervical cancer (Table 2).

Out of 530 total respondents, 371 (70%) had positive attitudes with a 95% CI [65.9, 74] and a mean of 1.7000, and a standard deviation of ± 0.45869 (Figure 2).

This study identified that more than half 291 (54.9%) and nearly three-fourths 387 (73%) of the daughters of parents were willing to regularly consult a medical doctor for cervical cancer screening and those daughters with multiple sex partners would be at higher risk for cervical cancer, respectively. The majority of parents of daughters disagreed that long-term use of contraceptive pills could cause cervical cancer and agreed that the use of condoms could reduce the risk of cervical cancer 413 (77.9%) and 386 (72.8%), respectively. More than half 370 (69.8%) of the respondents would not consult a medical doctor in case of abnormal bleeding between menstrual periods, while 386 (72.8%) parents of daughters agreed that HPV vaccination can prevent cervical cancers. The majority were willing to get a Pap smear test and agreed that a regular Pap smear test helps in the early detection of cervical cancer 434 (81.9%) and 483 (91%), respectively. The majority of respondents 467 (88.1) disagreed to pay for a Pap smear test and 463 (87.4%) respondents agreed that the government should provide free screening programs to reduce cervical cancer prevalence. The majority 434 (81.9%) of parents of daughters disagreed that the HPV vaccine might cause short-term problems, like fever or discomfort (Table 3).

Out of 530 total respondents, 450 (84.9%) parents accepted to vaccinate their daughter for HPV vaccination with a 95% CI [81.9, 87.9] and a mean of 1.1509 and a standard deviation of ± 0.35833 (Figure 3) and obstacles (Figure 4).

Out of 530 total respondents, 450 (84.9%) parents accepted to vaccinate their daughter for HPV vaccination with a 95% CI [81.7, 88.1] and a mean of 1.1509, and a standard deviation of ± 0.35833 (Figure 5).

The majority of the parents of daughters had heard of HPV and had heard of the HPV vaccine 467 (88.1%) and 450 (84.9%), respectively. The majority 450 (84.9%) of the study respondents were willing to accept vaccinating their daughter with HPV vaccination and 32 (6%) had reasons for being unwilling to take the HPV vaccine, such as being worried about the price, and the majority 482 (90.9%) of respondents did not accept that they should pay for the HPV vaccination by themselves (Table 4).

### 3.2. Factors Associated with the Acceptance of Human Papillomavirus Vaccine among Parents of Daughters

Multivariable analysis was used to control potential confounders. Having parents of daughters of the male sex, having only one daughter, having daughter(s) who attended a government school, having poor knowledge, and having a negative attitude were discovered to have a substantial relationship with acceptance of the human papillomavirus vaccine with a *p*-value < 0.05.

In this study, parents of daughters of the male sex were 59.3% less likely to accept HPV vaccination for their daughters as compared to the female sex (AOR: 0.407; 95%CI: 0.221, 0.748), parents who had only one daughter were 2.122 times more likely to accept HPV vaccination for their daughters as compared to those who had more than one daughter (AOR: 2.122; 95%CI: 1.221, 3.685), parents of daughters who attended a government school were 52.4% less likely to accept HPV vaccination for their daughters as compared to those whose daughter(s) attended a private school (AOR: 0.476; 95%CI: 0.263, 0.861), parents of daughters who had poor knowledge were 46.8% less likely to accept HPV vaccination for their daughters as compared to good knowledge (AOR: 0.532; 95%CI: 0.293, 0.969), and those parents of daughters who had a negative attitude were 46% times less likely to accept HPV vaccination for their daughters as compared to positive attitudes (AOR: 0.540; 95%CI: 0.299, 0.977) (Table 5).

## 4. Discussion

This study was conducted on parents of daughters attending school to ascertain parental awareness, knowledge, attitudes, and views regarding HPV infection, cervical cancer, and HPV vaccination as well as the acceptance of the HPV vaccine. The major purpose of this study was to determine parental awareness, acceptance, and associated factors of the human papillomavirus vaccine in the Hadiya zone, southern Ethiopia.

The main finding of this study revealed that, despite the views of some men and parents sending their daughters to government schools, parents indicated they were willing to vaccinate their children. This study finding is also almost nearly two times more than the study conducted in Hong Kong [23], and 1.12 times more than that conducted in Lagos, Nigeria [24].

According to the findings, the overall acceptance of HPV vaccinations for their daughters was 84.9%, 3.6 times higher than the study conducted among male baccalaureate students in Hong Kong [1], 1.1 times higher than the study conducted in Gondar town, northwest Ethiopia [22], 2.12 times higher than the study conducted in the United States [14], 1.24 times more than the study conducted in a premier medical school in India [9], and 1.12 times more than the study conducted in the Bench-Sheko zone, southwest Ethiopia [2].

These findings are almost 1.13 times lower than the study conducted in Indonesia [25], 1.12 times lower than in Addis Ababa, Ethiopia [4],1.14 times lower than in Lagos, Nigeria [26], 1.15 times lower than in the Kilimanjaro region, Tanzania [27], and 1.2 times less than in rural areas of Negeri Sembilan, Malaysia [28], but consistent with the previous study conducted at the University of KwaZulu-Natal, South Africa [29]. The ongoing national HPV vaccination program in Ethiopia may have contributed to the study’s high vaccine acceptance rate. The variation in acceptance rate from the Hadiya zone, southern Ethiopia study could be attributable to changes in participant socio-demographic characteristics and the sort of population employed by these studies.

The current national drive in Ethiopia for HPV vaccination may have contributed to the high vaccine acceptance rate in the study. During the data collection for this study, the parents were informed of the campaign-based school-based immunization program, and they decided on the vaccine. The high acceptability of the HPV vaccine in our study could be viewed as a chance to expand the country’s school-based HPV vaccination program. The study’s different acceptance rates in Addis Ababa, Gondar, and the Bench-Sheko zone could be partially attributed to the respondents’ varied sociodemographic makeup.

Given that the nation has started a school-based vaccination program for these targeted girls, this shows that Ethiopia’s high parental desire to vaccinate their daughters is good news and presents an excellent opportunity as well. Because parents have a significant role in their daughters’ decision to receive the vaccine or not, a greater parental acceptance to have their daughters receive the HPV vaccine will facilitate the vaccination campaign by making more girls available for vaccination.

In this study, parents of daughters of the male sex were 59.3% less likely to accept HPV vaccination for their daughters as compared to the female sex (AOR: 0.407; 95%CI: 0.221, 0.748), which was almost similar to the study conducted at a premier medical school in India [9], in Hong Kong [1], in rural areas of Negeri Sembilan, Malaysia [28], and the Bench-Sheko zone, Southwest Ethiopia [2]. This could be because women are more likely to know about vaccine availability, constitute the target group for immunization and the catch-up campaign, and women may also be more willing to accept the vaccine and suggest it to others.

Those parents of daughters who had only one daughter were 2.12 times more likely to accept HPV vaccination for their daughter as compared to the number of parents with more than one daughter (AOR: 2.122; 95%CI: 1.221, 3.685). This study found this result to be 1.57 times more than that of the study conducted in Shenzhen, China [11]. This could be because in sum, many parents are willing but have not vaccinated their daughters due to logistical barriers.

Parents of daughters who attended a government school were 52.4% less likely to accept HPV vaccination for their daughters as compared to those whose daughters attended a private school (AOR: 0.476;95%CI: 0.263,0.861); this could be because private schools are funded wholly or partly for the vaccination of daughters for human papillomavirus by a private body.

Parents of daughters who had poor knowledge were 46.8% less likely to accept HPV vaccination for their daughters as compared to good knowledge (AOR: 0.532; 95%CI: 0.293, 0.969), This study finding was consistent with the study conducted in Gondar town, northwest Ethiopia [22], almost 1.5 times more than the studies conducted in the Bench-Sheko zone, southwest Ethiopia [2], and this study finding was in contrast with the study finding in Arab communities [15].

This study finding is similar to the study conducted in Addis Ababa, Ethiopia [4], at a teaching hospital in Kuala Lumpur [30], in the Kilimanjaro region, Tanzania [27], but this study finding is also 2.82 times more than the study conducted in Lagos, Nigeria [24].

These disparities may have been brought about by the gap in respondents’ sociodemographic characteristics. This, in our opinion, demonstrates how parents’ acceptance to vaccinate their daughters is influenced by their knowledge of HPV vaccination. Therefore, raising parental knowledge is crucial, and programs should be developed specifically for parents without a formal education.

Parents of daughters who had negative attitudes were 46% times less likely to accept HPV vaccination for their daughters as compared to positive attitudes (AOR: 0.540;95%CI: 0.299,0.977). Parental positive attitude towards HPV vaccine was significantly associated with parental acceptance to vaccinate their daughters. This is supported by other study findings [2], e.g., in Gondar town, northwest Ethiopia [22]. This shows that more parents had a positive attitude toward the HPV vaccine despite their lack of understanding, and if vaccination services are offered, they would be willing to vaccinate their daughters.

## 5. Conclusions

This study found that there was a high level of parental acceptance to immunize their daughters against the human papillomavirus in the study area. Parents of daughters of the male sex, who had multiple daughters, whose daughters attended government schools, who had poor knowledge, and who had a negative attitude were discovered to have a substantial relationship with acceptance of the human papillomavirus vaccine. Additionally, the parents’ attitudes and knowledge about the HPV vaccine were significant in determining their intentions to vaccinate their daughters.

It is advised that there is a need to provide national educational campaigns about the HPV vaccine to the public given that the HPV vaccine is not included in the national immunization schedule. Authorities in high-risk areas for cervical cancer incidence should plan and implement strategies by providing health information regarding the human papillomavirus vaccination, with an emphasis on raising community awareness.

The regional and zonal health offices in Ethiopia’s Ministry of Health should develop health information regarding HPV vaccination with a focus on raising community awareness, especially among parents who lack formal education. It is essential to raise community awareness sustainably to secure the long-term acceptability of HPV vaccination. Therefore, it is essential to improve awareness and acceptance of cervical cancer as well as its prevention in the community.

## 6. Limitation

To demonstrate the current parental acceptance to vaccinate their daughters in response to the vaccine campaign being run by the Ethiopian MOH and Zonal Health Department, a community-based study with a representative sample of parents from Ethiopia’s rural areas may be feasible. Additionally, it is the first study from a zonal level in Ethiopia to fully characterize parental understanding and acceptance of the HPV vaccination. Since the study is cross-sectional, a causal relationship cannot be shown; nonetheless, recollection and social desirability bias may potentially have an impact.

## Figures and Tables

**Figure 1 vaccines-10-01988-f001:**
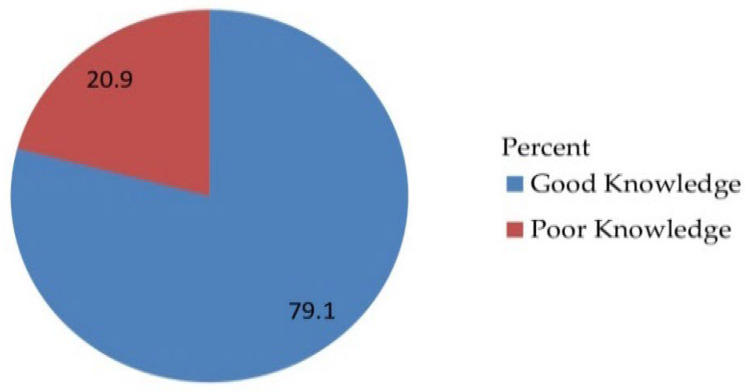
Knowledge-related questionnaire of the respondents on awareness, acceptance, and associated factors of human papillomavirus vaccine among parents of daughters in Hadiya zone, southern Ethiopia, 2021 (n = 530).

**Figure 2 vaccines-10-01988-f002:**
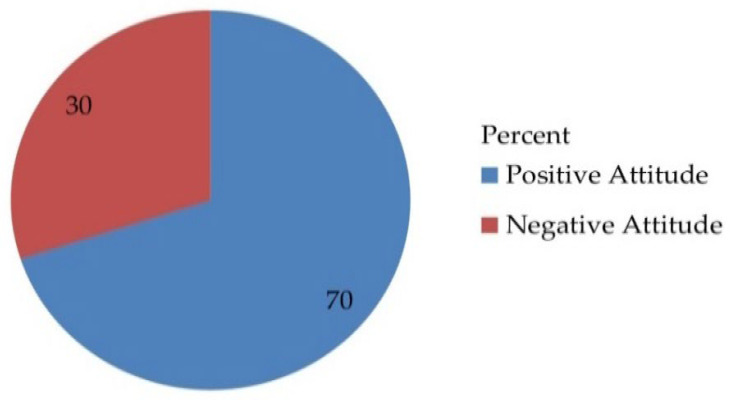
Attitude-related questionnaire of the respondents on awareness, acceptance, and associated factors of human papillomavirus vaccine among parents of daughters in Hadiya zone, southern Ethiopia, 2021 (n = 530).

**Figure 3 vaccines-10-01988-f003:**
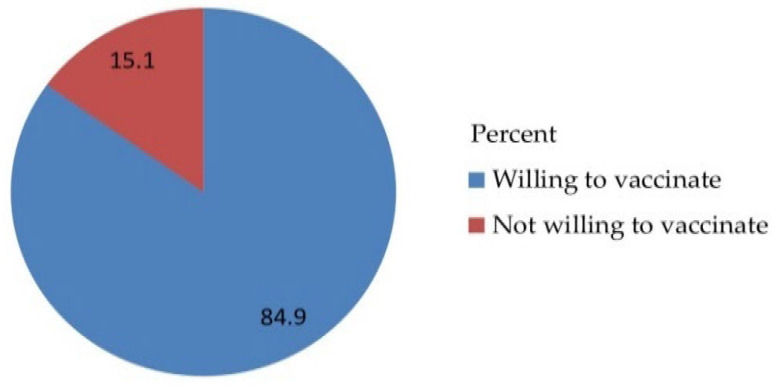
Acceptance-related questionnaire of the respondents on awareness, acceptance, and associated factors of human papillomavirus vaccine among parents of daughters in Hadiya zone, southern Ethiopia, 2021 (n = 530).

**Figure 4 vaccines-10-01988-f004:**
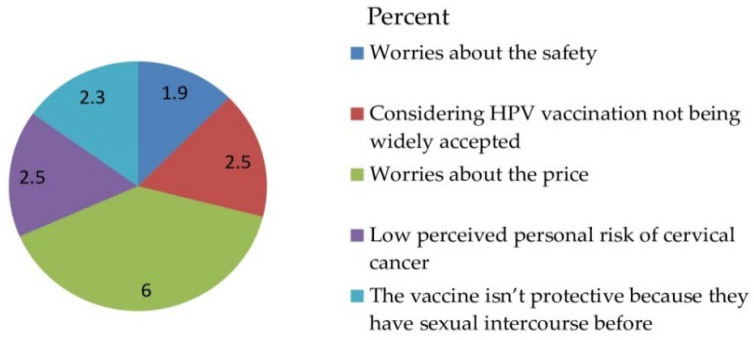
Obstacles to the acceptance-related questionnaire of the respondents on awareness, acceptance, and associated factors of human papillomavirus vaccine among parents of daughters in Hadiya zone, southern Ethiopia, 2021 (n = 530).

**Figure 5 vaccines-10-01988-f005:**
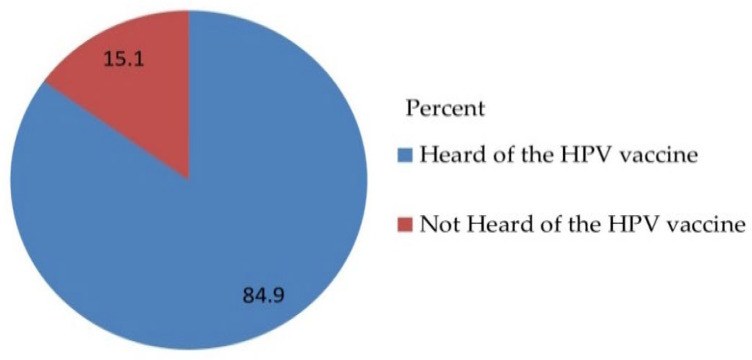
Awareness-related questionnaire of the respondents on awareness, acceptance, and associated factors of human papillomavirus vaccine among parents of daughters in Hadiya zone, southern Ethiopia, 2021 (n = 530).

**Table 1 vaccines-10-01988-t001:** Sociodemographic-related characteristics of the respondents on awareness, acceptance, and associated factors of human papillomavirus vaccine among parents of daughters in Hadiya zone, southern Ethiopia, 2021 (n = 530).

Variables	Categories	n (%)
Sex	Male	207 (39.1)
	Female	323 (60.9)
Age in years	18–29	80 (15.1)
	30 to 39	323 (60.9)
	≥40	127 (24)
Residence	Rural	176 (33.2)
	Urban	354 (66.8)
What is your marital status now?	Single	69 (13)
	Married	437 (82.5)
	Never married	24 (4.5)
Religion	Orthodox	86 (16.2)
	Muslim	64 (12.1)
	Protestant	365 (68.9)
	Catholic	15 (2.8)
Educational status	Unable to read and write	16 (3)
	Primary and secondary education	96 (18.1)
	Certificate	32 (6)
	Diploma	64 (12.1)
	Degree and above	322 (60.8)
Occupation	Civil servant	354 (66.8)
	Self-employed	80 (15.1)
	Merchant	80 (15.1)
	Housewife	16 (3)
Ethnicity	Hadiya	389 (73.4)
	Kembata	50 (9.4)
	Amhara	8 (1.5)
	Tigre	16 (2.1)
	Gurage	10 (1.9)
	Oromo	9 (1.7)
	Wolaita	16 (3)
	Gurage	32 (6)
	Silte	16 (3)
	Others *	41 (5.4)
Number of daughters	One	112 (21.1)
	More than one	418 (79.8)
School type	Government School	175 (33)
	Private school	355 (67)
Information about HPV infection	Yes	318 (60)
	No	212 (40)
Sources of information about HPV infection (more than one possible answer possible)	TV or Radio	144 (45.3)
	Health workers	78 (24.5)
	Their friend	70 (22)
	Others **	26 (8.2)
Monthly income	≤ETB 4999 (≤USD 95.24)	239 (45.1)
	>ETB 4999 (>USD 95.24)	291 (54.9)

Where: TV; television, ETB; Ethiopian total birr, USD; United States dollars * = 23 Alaba, 6 Sidama, 12 Dawro, ** = 12 Internet, 5 email messages, 7 groceries, and 2 newspapers.

**Table 2 vaccines-10-01988-t002:** Knowledge-related questionnaire of the respondents on awareness, acceptance, and associated factors of human papillomavirus vaccine among parents of daughters in Hadiya zone, southern Ethiopia, 2021 (n = 530).

Variables	Categories	n (%)
Have you ever heard about cervical cancer?	Yes	418 (78.9)
	No	112 (21.1)
Does cervical cancer only occur in females?	Yes	499 (94.2)
	No	31 (5.8)
Cervical cancer at an early stage produces no signs or symptoms	Yes	111 (20.9)
	No	419 (79.1)
Cervical cancer is a fast-growing cancer	Yes	144 (27.2)
	No	386 (72.8)
If detected early, is cervical cancer curable?	Yes	418 (78.9)
	No	112 (21.9)
Getting a Pap test helps early detection of cervical cancer	Yes	371 (70)
	No	159 (30)
Human papillomavirus is the main cause of cervical cancer	Yes	387 (73)
	No	143 (27)
Human papillomavirus is very common in women younger than 30 years	Yes	435 (82.1)
	No	95 (17.9)
Human papillomavirus vaccination is available for girls aged 9 to 14 years	Yes	419 (79.1)
	No	111 (20.9)
Cervical cancer risk can be reduced by HPV vaccination	Yes	434 (81.9)
	No	96 (18.1)
Do you know that HPV vaccination should be received before sexual debut?	Yes	338 (63.8)
	No	192 (36.7)
Do you know that HPV infection is a sexually transmitted infection?	Yes	307 (57.9)
	No	223 (42.1)
Do you know that the persistent infection of high-risk HPV could lead to cervical cancer?	Yes	434 (81.9)
	No	96 (18.1)
Do you know the symptoms of cervical cancer?	Yes	386 (72.8)
	No	144 (27.2)
Do you know the time duration that it takes for abnormal cervical cells to turn into cancerous cells?	Yes	80 (15.1)
	No	450 (84.9)
70% of cervical cancer is caused by HPV types 16 and 18	Yes	112 (21.1)
	No	418 (78.9)
Do you know the most common and effective treatment to cure cervical cancer?	Yes	159 (30)
	No	371 (70)

Where: HPV, human papillomavirus; symptoms of cervical cancers: vaginal bleeding after intercourse, between periods, or after menopause; watery, bloody vaginal discharge that may be heavy and have a foul odour; pelvic pain or pain during intercourse.

**Table 3 vaccines-10-01988-t003:** Attitude-related questionnaire of the respondents on awareness, acceptance, and associated factors of human papillomavirus vaccine among parents of daughters in Hadiya zone, southern Ethiopia, 2021 (n = 530).

Variables	Strongly Disagree		Disagree		Neutral		Agree		Strongly Agree	
	n	%	n	%	n	%	n	%	n	%
Are you willing to regularly consult a medical doctor for cervical cancer screening?	175	33	16	3	48	9.1	291	54.9	0	0
Those with multiple sex partners will be at higher risk for cervical cancer	64	12.1	15	2.8	64	12.1	371	70	16	3
Long-term use of contraceptive pills could cause cervical cancer	112	21.1	296	55.8	5	0.9	90	17	27	5.1
The use of condoms could reduce the risk of cervical cancer	16	3	15	2.8	80	15.1	64	12.1	355	67
HPV vaccination can prevent cervical cancers	48	9.1	32	6	64	12.1	80	15.1	306	57.7
Will you consult a medical doctor in case of abnormal bleeding between menstrual periods?	274	51.7	16	3	80	15.1	80	15.1	80	15.1
Regular Pap smear test helps early detection of cervical cancer	16	3	16	3	15	2.8	355	67	128	24.2
Are you willing to get a Pap smear test?	32	6	32	6	32	6	111	20.9	323	60.9
Are you willing to pay for a Pap smear test?	64	12.1	64	12.1	339	64	32	6	31	5.8
Government should provide free screening programs to reduce cervical cancer prevalence	32	6	31	5.8	7	1.3	364	68.7	96	18.1
The HPV vaccine might cause short-term problems, like fever or discomfort	16	3	32	6	386	72.8	48	9.1	48	9.1

**Table 4 vaccines-10-01988-t004:** Acceptance- and awareness-related questionnaire of the respondents on awareness, acceptance, and associated factors of human papillomavirus vaccine among parents of daughters in Hadiya zone, southern Ethiopia, 2021 (n = 530).

Variables	Categories	n (%)
Heard of HPV	Yes	467 (88.1)
	No	63 (11.9)
Heard of the HPV vaccine	Yes	450 (84.9)
	No	80 (15.1)
Are you willing to vaccinate your daughter for HPV?	Yes	450 (84.9)
	No	80 (15.1)
The reasons for being unwilling to take the HPV vaccine (more than one possible answer possible (obstacles))	Worried about the safety	10 (1.9)
	Considers HPV vaccination not widely accepted	13 (2.5)
	Worried about the price	32 (6)
	Low perceived personal risk of cervical cancer	13 (2.5)
	The vaccine is not protective because they have had sexual intercourse before	12 (2.3)
Accept that they pay for HPV vaccination by themselves	Yes	48 (9.1)
	No	482 (90.9)

**Table 5 vaccines-10-01988-t005:** Multivariable and bivariate analyses of factors associated with the acceptance of human papillomavirus vaccine among parents of daughters in Hadiya zone, southern Ethiopia, 2021 (n = 530).

Variable	Acceptability of the HPV Vaccination				
	No	Yes	COR (95%CI)	AOR (95%CI)	*p*-Value
Sex					
Male	48 (60)	159 (35.3)	0.364 (0.224, 0.539)	0.407 (0.221, 0.748) *	0.004
Female	32 (40)	291 (64.7)	1	1	
Number of daughters					
One	32 (40)	80 (17.8)	0.324 (0.195, 0.539)	2.122 (1.221,3.685) *	0.008
More than one	48 (60)	370 (82.2)	1	1	
School type					
Government School	48 (60)	127 (28.2)	0.262 (0.160, 0.429)	0.476 (0.263, 0.861) *	0.014
Private school	32 (40)	323 (71.8)	1	1	
Knowledge					
Poor knowledge	32 (40)	79 (17.6)	0.319 (0.192, 0.531)	0.532 (0.293, 0.969) *	0.039
Good knowledge	48 (60)	371 (82.4)	1	1	
Attitude					
Negative Attitude	32 (40)	127 (28.2)	0.590 (0.361, 0.965)	0.540 (0.299, 0.977) *	0.042
Positive Attitude	48 (60)	323 (71.8)	1	1	

Where: COR; crude odds ratios, AOR; adjusted odds ratios, 1 = Reference, * shows the variable significant at *p*-value < 0.05 in multi-variable analysis.

## Data Availability

The datasets used and/or analyzed during the current research are available upon request from the corresponding author.

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
