# Peer review of "Awareness, Acceptance, and Associated Factors of Human Papillomavirus Vaccine among Parents of Daughters in Hadiya Zone, Southern Ethiopia: A Cross-Sectional Study"

_vaccines, 2022, doi:10.3390/vaccines10121988_

Round 1

Reviewer 1 Report

This is an informative article on HPV vaccine awareness and knowledge among parents of age-eligible females in the Hadiya Zone of Southern Ethiopia. I have some comments that may help strengthen this article.

Introduction: 

You could also mention oropharyngeal cancers as well, given that HPV causes about 70% of those cancers. 

Page 2, line 44: You may want to clarify this sentence to add "if an HPV infection goes untreated without removal of precursor lesions, it can lead to cancer." The way the sentence reads now, it seems to imply there is treatment for HPV infection, which doesn't currently exist. We're limited to a management strategy involving screening and treatment to reduce cancer risk. 

Page 2, line 51: This paragraph is a bit confusing and seems somewhat repetitive. I think you could leave out the number of new cases and deaths since it's mentioned in the first paragraph. You could start at line 54 ("Sub-Saharan Africa has the highest incidence and mortality rates..."

Page 2, line 66: "Adolescence" may be a more appropriate word in this sentence because readers may interpret the word childhood differently.  

Page 3, line 115: What are the demographic characteristics of this area that would be helpful for readers outside of Ethiopia to know? Is this a poorer area, dominated by one specific religion, or is there extremely limited healthcare availability, etc.?

Methods:

Page 3, line 130: The information in this sentence seems to be repeated from the line above it. 

Page 3, line 144: I'm a little confused about the sampling procedure. Were eligible households recruited through area schools? You may want to consider clarifying here. 

Page 5, line 208: 

Given the stratified sampling design and probability proportional to size sampling, your frequencies should be weighted and SPSS survey procedures used to analyze the data. Were you intending to generalize your findings to all parents of daughters in this region? Here is some information on using SPSS for complex survey data analysis: https://digitalcommons.wayne.edu/cgi/viewcontent.cgi?article=3253&context=jmasm

Page 5, line 213: Technically, you should use the term "multivariable" instead of "multivariate" when you have one dependent variable in the model. 

Page 5, line 216: What does this number mean?

Results: page 5, line 233: What was the income range?

Table 2: Given that cervical cancer is an outcome of HPV infection, it may be more appropriate to state "does cervical cancer only occur in females?" Was this how it was worded in the survey?

Page 8, line 278: "parents of daughters agreed that long term use of contraceptive pills...." Should this be changed to "disagree" to match table 3's results?

Page 11, line 322: it would be simpler to state "male parents/guardians" rather than the current wording, given that readers will know from the methods section that parents/guardians were the ones being surveyed. 

Page 11, line 327: It would be simpler to say "having only one daughter." 

Table 5: What does COR stand for?

Discussion: Page 12, lines 345 - 347: These sentences are basically repeated from the Introduction, so there's no need to restate this information. 

Page 12, line 349: I'm not sure this sentence actually conveys what you mean. Instead, do you mean this? "Despite the views of some men and parents sending their daughters to government-schools, parents indicated they were willing to vaccinate their children. "

Page 12, line 355: I think you could just report the percentage here. 

Page 12, line 367: Do you mean "acceptance" instead of "uptake" given the focus of the survey on examining vaccine awareness and acceptance? 

Page 12, line 385: Women may also be more aware of cervical cancer and the consequences of not being vaccinated, such as being diagnosed with cervical cancer. 

page 13, line 405: How well do these survey participants represent the region overall? They seem to be more highly educated, so I'm wondering if that influenced the results, including why knowledge and awareness was higher in this study compared to countries of similar status. 

Author Response

For Reviewer 1:

Introduction: HPV causes about 70% of those cancers (90-95% of the oropharyngeal cancers should be caused by HPV type 16 which is the most common it should be added, revised, and updated).

Page 2, line 44: "if an HPV infection goes untreated without removal of precursor lesions, it can lead to cancer." Should be added, revised, and updated

Page 2, line 51: Repetitive removed, updated, and revised

Page 2, line 66: Adolescence: added, revised, and updated. 

Page 3, line 115: the demographic characteristics of the study setting would be added, updated, and revised.

Methods:

Page 3, line 130: updated and revised (if am not mistaken).

Page 3, line 144: regarding the sampling procedure, the preliminary assessment survey was done before going to actual data collection; which means how many total numbers of daughters attend primary school (private and public), from the zonal educational office, we also communicate health extension workers(there is a family household registration book) adolescents with age category 9-14 years, then finally we will make mark households selected randomly and finally data should be collected by interview administered questions through households by asking parents of daughters until we reach required sample size, but not the eligible households recruited through area schools. Revised and updated some.

Page 5, line 208: we are intending to generalize our findings to all parents of daughters in the Hadiya region and also beyond the region to vaccinate their daughters, those interested in research to conduct the study with similar topics with more strong study design and for the concerning authorities to emphasize human papillomavirus vaccine importance in regarding prevention and control of cervical and others cancers should happen substantially without it.

Page 5, line 213: multivariable replaced instead of multivariate: revised and updated.

Page 5, line 216: the number removed and significant finding acceptable

Results: page 5, line 233: the household income should be asked by open-ended questionnaire form and categorized based on the study conducted with a similar setup or standard living conditions with our region of a zone with a similar study design, based on this study it have two range categories those parents of daughters with income greater >4999 ETB (>95.24 USD) consider as high-income categories and the reverse is also true. 

Table 2: During survey data, collection time orientation or training is given for the data collectors concerning symptoms like the asked some vaginal bleeding after intercourse, between periods, or after menopause; Watery, bloody vaginal discharge that may be heavy and a foul odor; Pelvic pain or pain during intercourse: revised and updated

Page 8, line 278: agreed replaced with disagree: corrected

Page 11, line 322: parents/guardians: revised and corrected

Page 11, line 327: having only one daughter: revised and replaced.

Table 5: revised and updated

Discussion:

Page 12, lines 345 - 347: repeated words removed: updated and revised.

Page 12, line 349: "Despite the views of some men and parents sending their daughters to government schools, parents indicated they were willing to vaccinate their children" this word was replaced and updated.

Page 12, line 355: the report should be corrected into a percentage.

Page 12, line 367: uptake replaced into acceptance: corrected and revised

Page 12, line 385: Yes of course why the prevalence of the case of cervical can be high.

Page 13, line 405: Yes and unfortunately in this survey majority 60.8% of parents of daughters with the educational status degree and above.

Reviewer 2 Report

The paper presents the results of a cross-sectional study of knowledge and beliefs about HPV vaccination in parents of daughters in the Hadiya Zone in Southern Ethiopia. The data is very interesting, however the work requires some adjustment and clarification of a few issues.

1. The introduction is too long, makes the same statements over and over, and some important information is missing.

First, not every HPV infusion leads to cancer. Relevant data showing the course of infection should be shown - possibility of elimination by the body or tumor development (preceded by appropriate precancerous conditions)

Second, there is a lack of information on additional factors predisposing to the development of HPV-dependent neoplasms.

Third, there is no information about other than cervical cancer - HPV-dependent cancers.

2. According to Authors: "Almost half 275(51.9%) respondents 232 were monthly income >4999 ETB (≤95.24USD) (Table 1)." 

>4999 ETB is rather more than ≤95.24USD.

There is no Table 1 in the text. First table is "Table 2". Additionally it is very poor readable

3. The way the data are presented in the charts is wrong. Total and percentage values cannot be stacked side by side. A pie chart would be better, although the data is contained in the text and tables, so is it worth repeating?

4. Whether the questionnaire used was original or was it used in other studies? If it wasn't found in other studies, how was it different from previous questionnaires?

5. Too little space in the discussion was devoted to comparing with other countries and discussing where the potential differences come from.

Author Response

Thank you for the comments

  1. The introduction is some narrowed, the statements repeated were removed, and the necessary important information was added: revised and updated.

We tried adding additional factors predisposing to the development of HPV concerning those mothers with a history of HIV: updated and revised

The information other than cervical cancer especially concerning oropharyngeal cancer: updated and revised

  1. Income in Table 1: updated and revised

Table 2 was replaced into the first table and the poor readable way some modified

  1. The way the data are presented was changed into a pie chart. I don't think the data presented through the table with some notation should be written above the table with high frequency: revised and updated.
  2. The questionnaire used was developed by reading different literature, which is used in other studies and not original.
  3. The potential differences: updated and revised

Reviewer 3 Report

This paper will be important for the first study from a zonal level in Ethiopia to characterize parental understanding and acceptance of the HPV vaccination. Such work is attractive to a broader audience across the world. The data contributes to preventing cervical cancer in Ethiopia and other countries.

In my opinion, the paper is accepted for publication after minor revision.

1. Although the plan and current situation in Ethiopia is described in detail, it would be better to explain the standard of living in the Hadiya Zone of Ethiopia compared to other regions in the introduction or discussion.

Since we are not familiar with the environment of the " Hadiya Zone," it would be helpful to include information on differences between different regions, such as education level, household income, family structure, age of first childbirth, etc., to give a clearer picture of Ethiopia.

2. Some comparisons with other countries and regions are mentioned in the discussion regarding the acceptability of HPV vaccination among parents with daughters (lines 354 to 365), but it would be more helpful to add a discussion of the causes of the differences in the numbers, if possible.

3. There are few descriptions of adverse events related to the vaccine. It would be good if the extent of adverse events caused by HPV in Ethiopia could be described to the extent that they have been investigated.

4. I think table 2 on line 234 is a mistake for table 1. Please fix it.

Author Response

Thank you for the comments

  1. The standard of the living condition of the Hadiya Zone of Ethiopia should be added in the introduction: revised and updated.
  2. The causes of the differences in the numbers should be mentioned: revised and updated
  3. There is no further study conducted on adverse events related to the HPV vaccine in Ethiopia to add more. If we are not mistaken: a little bite and some modifications done.
  4. Table 2 replaced in table 1: updated and revised

For grammatical editing we tried to invite for different free sample language editors like papers true and we all tried to see our document ergrously.

Round 2

Reviewer 1 Report

The revisions do improve this manuscript. However, the pie charts need modification to remove the totals as a slice of the pie, since each slice added together should sum to the total. An alternative to a pie chart is a doughnut chart: https://support.microsoft.com/en-us/office/present-your-data-in-a-doughnut-chart-0ac0efde-34e2-4dc6-9b7f-ac93d1783353#:~:text=Select%20the%20data%20that%20you,%2C%20Layout%2C%20and%20Format%20tabs.

Page 8, line 292: I would reword this sentence to the following: The majority 413 (77.9%) and 386 (72.8%) of parents of  daughters disagreed that long-term use of contraceptive pills could cause cervical cancer and agreed that use of condoms could reduce the risk of cervical cancer, respectively.

Page 8, line 295: I suggest replacing the phrase "were doesn't" with "wouldn't" for improved readability. 

Page 11, line 342: I suggest rewording this sentence to something like this: " ...parents having only one daughter were 2.122 times more likely to accept HPV vaccination for their daughters as compared to those having more than one daughter (AOR: 2.122; 95%CI: 1.221,3.685),...

Author Response

we tried to adress all the comments

thank you!

Reviewer 2 Report

All reviewers comments have been addressed. 
No further comments.

Author Response

thank you